# The Relationship between Patient Self-Reported, Pre-Morbid Physical Activity and Clinical Outcomes of Inpatient Treatment in Youth with Anorexia Nervosa: A Pilot Study

**DOI:** 10.3390/nu16121889

**Published:** 2024-06-15

**Authors:** Martina Pech, Christoph U. Correll, Janine Schmidt, Almut Zeeck, Tobias Hofmann, Andreas Busjahn, Verena Haas

**Affiliations:** 1Department of Child and Adolescent Psychiatry, Charité–Universitätsmedizin Berlin, Corporate Member of Freie Universität Berlin and Humboldt–Universität zu Berlin, 13353 Berlin, Germany; christoph.correll@charite.de (C.U.C.); janine-d.schmidt@charite.de (J.S.); 2Donald and Barbara Zucker School of Medicine at Hofstra/Northwell, Hempstead, NY 11549, USA; 3Department of Psychiatry, The Zucker Hillside Hospital, Glen Oaks, NY 11004, USA; 4Department of Psychosomatic Medicine and Psychotherapy, Center for Mental Health, Faculty of Medicine, University of Freiburg, 79104 Freiburg im Breisgau, Germany; almut.zeeck@uniklinik-freiburg.de; 5Charité Center for Internal Medicine and Dermatology, Department for Psychosomatic Medicine, Charité-Universitätsmedizin Berlin, Corporate Member of Freie Universität Berlin, Humboldt-Universität zu Berlin and Berlin Institute of Health, 12203 Berlin, Germany; tobias.hofmann@charite.de; 6Department of Psychosomatic Medicine and Psychotherapy, DRK Kliniken Berlin Wiegmann Klinik, 14050 Berlin, Germany; 7Health TwiSt GmbH, 13125 Berlin, Germany; abusjahn@healthtwist.de

**Keywords:** premorbid physical activity, adolescents, accelerometry, eating disorders, clinical outcome, semi-structured interview

## Abstract

Links between premorbid physical activity (PA) and disease onset/course in patients with anorexia nervosa (AN) remain unclear. The aim was to assess self-reported PA as a predictor of change in percent median BMI (%mBMI) and length of hospital stay (LOS). Five PA domains were assessed via semi-structured interview in adolescents with AN at hospitalization: premorbid PA in school grades 1-6 (PA1-6); PA before AN onset (PA-pre) and after AN onset (PA-post); new, pathological motivation for PA (PA-new); and high intensity PA (PA-high). Eating disorder psychopathology was measured via the Eating Disorder Examination Questionnaire (EDE-Q), and current PA (steps/day) with accelerometry. PA1-6 was also assessed in healthy controls (HCs). Using stepwise backward regression models, predictors of %mBMI change and LOS were examined. Compared with 22 HCs (age = 14.7 ± 1.3 years, %mBMI = 102.4 ± 12.1), 25 patients with AN (age = 15.1 ± 1.7 years, %mBMI = 74.8 ± 6.0) reported significantly higher PA1-6 (median, AN = 115 [interquartile range IQR = 75;200] min vs. HC = 68 [IQR = 29;105] min; *p* = 0.017). PA-post was 244 ± 323% higher than PA-pre. PA1-6 was directly associated with PA-pre (*p* = 0.001) but not with PA-post (*p* = 0.179) or change in PA-pre to PA-post (*p* = 0.735). Lower %mBMI gain was predicted by lower baseline %mBMI (*p* = 0.001) and more PA-high (*p* = 0.004; r^2^ = 0.604). Longer LOS was predicted by higher PA-pre (*p* = 0.003, r^2^ = 0.368). Self-reported PA may identify a subgroup of youth with AN at risk of less weight gain and prolonged LOS during inpatient treatment for AN.

## 1. Introduction

Increased physical activity (PA) was recognized as early as 1873 as part of one of the first descriptions of anorexia nervosa (AN) by William Gull [1]. Today, increased PA is regarded as a significant factor in the inception and persistence of AN [2], being associated with a longer duration of inpatient treatment [3] and higher rates of a chronic outcome [4]. Between 40–80% of patients with AN show increased PA [5,6], but PA levels vary in different phases of AN and treatment of AN [7]. 

In a mixed adult and adolescent patient group with AN, we previously showed that inpatient weight restoration was slowed down by high objectively assessed PA in the form of steps and time spent in activities with moderate intensity [7]. Accelerometry is not currently incorporated into the routine clinical assessment of patients with AN, and the mode of interpreting accelerometrically assessed PA data in patients with AN has not been standardized. 

A qualitative interview study of the role of exercise across the lifespan in 24 females with a current or previous diagnosis of AN reported strong premorbid interest in exercise in 91.7% and a transformation into compulsive exercise behaviors and beliefs with the onset of AN in 87.5% of the sample [8]. In a structured interview study measuring PA during five different time frames (ages 8, 13, 18, 23, and 12 months before the interview) adult women with AN and BN were more physically active than healthy controls (HCs) from early adolescence onwards and prior to the onset of AN [9]. 

However, subjectively recalled autobiographical data on PA have so far not been combined with direct accelerometric information. Some studies in patients with AN have combined objective and subjective methods of quantifying PA, yielding heterogeneous results, suggesting both overreporting [10] and underreporting [11] of PA as well as no correlation between objectively and subjectively assessed PA [12]. Rizk et al. proposed a model for the development of abnormally increased PA in AN [13], but the timing of PA alterations in the development and course of AN deserves a more differentiated look. 

Therefore, the aims of this study were to (i) develop a short, semi-structured interview covering different domains of PA in order to assess patient-reported premorbid and current PA in adolescents with AN; (ii) apply the interview to a group of adolescent inpatients with AN and to HCs; (iii) evaluate interrater variability of the interview; and (iv) assess the potential of the interview to identify patients at risk of an unfavorable clinical outcome, i.e., lower gains in percent median BMI (%mBMI) during inpatient treatment of AN and/or extended inpatient length of stay (LOS). Based on prior data [2,3,4,5,6,7,8,9,10,11,12,13,14,15,16,17], we hypothesized that PA would increase from premorbid to morbid phases of AN and that increased PA would be associated with poorer outcomes.

## 2. Materials and Methods

Adolescents aged 12–18 years old with AN admitted to the Department of Child and Adolescent Psychiatry at the Charité–Universitätsmedizin Berlin for inpatient treatment between March 2016 and June 2018 were consecutively approached for participation in the study. This group represents a subgroup of a cohort described earlier [14]. Patient inclusion criteria were a diagnosis of AN according to ICD-10 (International Statistical Classification of Diseases and Related Health Problems, 10th Revision) of the restrictive, binge–purge, or atypical type. Exclusion criteria were a diagnosed psychotic episode or any somatic condition (e.g., hemiparesis) other than AN with a potential effect on PA. Information about the duration of the illness, psychiatric comorbidities, and psychotropic medications at the beginning of inpatient treatment were retrieved from medical records. Between 2017 and 2018, 22 age-matched female healthy controls, consisting of family and friends of clinical staff, were recruited. Exclusion criteria were any known major medical or psychiatric disease or any condition with a significant influence on PA. HCs were screened with the Sick, Control, One stone (14 lbs./6.5 kg), Fat, Food (SCOFF) questionnaire to exclude participants with eating disorders [18]. All participants aged 18 provided written informed consent, with minors providing written informed assent, and their legal guardians providing written informed consent for participation in the study, which was approved by the ethics committee of the Charité–Universitätsmedizin Berlin (Identification code: EA2/034/14; date of approval 24 June 2014). 

### 2.1. Anthropometry

The body weight of all patients was measured to the nearest 0.1 kg using a chair scale (MCB300K100M, KERN & Sohn GmbH, Balingen, Germany) and height to the nearest 0.5 cm using a stadiometer (Seca 220 Stadiometer, Vogel & Halke, Hamburg, Germany). Measurements took place in the morning between 7 and 8 AM after overnight fasting and in underwear. Body weight of the control group subjects was measured during the day (not fasted, because the appointments took place after school) using the same chair scale (MCB300K100M, KERN & Sohn GmbH, Balingen, Germany), and height was measured using the same stadiometer (Seca 220 Stadiometer, Vogel & Halke, Hamburg, Germany) [8]. Body Mass Index (BMI) was calculated as kg/m^2^, with BMI percentiles according to Kromeyer–Hausschild [19]. Percent median BMI was calculated as the lower border of the 50th BMI-for-age percentile for age and sex using the formula %mBMI = current body weight divided by median body weight × 100. Using %mBMI allows better distinguishing of the degree of underweight in individuals below the 1st BMI percentile [20]. Primary amenorrhea was defined as absence of menstruation after the conclusion of the 16th birthday, calling absent menstruation before this time point “no menarche until age 16”.

### 2.2. Assessment of PA and Eating Disorder Psychopathology

PA (steps/day) was measured in patients with AN within 21 ± 6 (range: 11–36) days after inpatient admission. Using a portable armband device (SenseWear™ PRO3 armband; BodyMedia, Inc., Pittsburgh, PA, USA), PA was assessed over a 3-day period (Friday to Sunday) [8,14,15]. As part of the inpatient treatment program, PA was limited. Patients with a weight below the 3rd BMI percentile had strict resting hours after mealtimes (60 min after the three main meals, 30 min after snacks). Patients were allowed to go for a 15 min daily walk outside and to move freely on the ward. To measure psychopathology related to AN, the German version of the Eating Disorder Examination Questionnaire (EDE-Q) [21,22] was used. Measurements of HCs took place while they stayed in their ambulatory environment. Otherwise, PA assessment was standardized and took place over a 3-day-period from Friday to Sunday as well.

### 2.3. Semi-Structured PA Interviews

Semi-structured PA interviews were conducted by the same interviewer (MP) with the patients with AN in person during inpatient treatment, and with the HCs in person (n = 10) or over the phone (n = 12). All participants were given identical instructions, and questions were asked in the same order. Information was obtained regarding PA from preschool until the current school grade, differing according to age. Every school grade was mentioned individually (“Did you do sports in first grade?” “What kind?” “How many times a week?” “For how long?”). Further, it was specified whether PA took place at or after school and in which setting. Activities that did not happen regularly were excluded, such as “often going to the pool with my friends” or “bicycle trips with the family on weekends”. If participants were unsure how many minutes their PA had lasted (e.g., 60 or 90 min) the lower amount was chosen. An excerpt of one sample interview can be found in the Appendix A.

### 2.4. PA Domains

PA domains were defined based on life events in chronological order that all participants had in common. All participants had finished elementary school.

(1)Premorbid PA in grades 1-6 (PA1-6): the median of weekly PA in minutes in grades 1-6.(2)PA before onset of AN (PA-pre): the last available information on PA in minutes/day before self-reported onset of AN. For example, when a person became ill in the second half of grade 9 and had previously, in the first half of grade 9, played basketball twice a week for 90 min each, PA-pre would be 180 min/week.(3)PA after onset of AN (PA-post): Patients were asked about a perceived change in PA at the time of the self-reported onset of AN and, where indicated, were asked to describe these changes in quality and quantity. The percent change of PA between pre-AN baseline and onset of AN was calculated for each participant.

If patients remained vague even after prompts (e.g., “Well, I swam a couple more laps.”), these answers were quantified as precisely as possible. Answers such as “I biked around the park three times each night” were also quantified using knowledge about local distances (1×around park = 6 km, 3× = 18 km). 

Patients were classified into two subgroups (*with* or *without* significant PA change) according to the following items since onset of AN relating to qualitative changes in PA. If one of the three criteria above was fulfilled, the patient was classified as having had a significant change in PA:(1)Equal to or more than 6 h/week of PA (yes/no); this cut-off is most commonly used in the literature to assess excessive PA in AN [6];(2)New onset and/or high intensity in PA (PA-high; yes/no), e.g., 13 slow sit-ups in 10 min or daily jogging of 900 m in 3.4 min (both examples are quotes from interview answers); and/or(3)New onset of PA motivation: PA (PA-new) utilized with the main purpose to lose weight, control shape, and/or to burn calories (yes/no).

Notes were taken throughout the interview and transcribed immediately afterwards. Two independent raters (MP, JS) assessed all classifications for each patient. Inconsistencies were resolved by consensus.

### 2.5. Clinical Outcome Parameters and Statistical Analysis

Analyses were conducted using R version 4.3.3 (29 February 2024). Co-primary outcome parameters were change of %mBMI during inpatient treatment and LOS. All analyses were two-sided with alpha = 0.05. All continuous data are presented as mean ± standard deviation (SD) if normally distributed, otherwise as median [25th/75th percentile] and range. Group differences were computed with either a *t*-test or Wilcoxon test accordingly. Categorical data were reported with absolute frequency and relative frequency in % and tested by Fisher’s exact test. Interviewer agreement was analyzed by Cohen’s Kappa. Due to the exploratory nature of outcome prediction in this study, we focused on a stepwise backward regression approach, after excluding co-linear variables assessed with Spearman’s rank correlation and retaining only significant independent variables in the final model. The underlying complete multivariable model as well as univariate analyses for all independent variables are reported in the Appendix A. Adjusted R-squared (r^2^) and the Akaike information criterion (AIC) values are reported as measures of model fit.

## 3. Results

Between 2014 and 2018, 106 patients were approached after hospitalization about participating in a study on PA in adolescent patients with AN, and 56 agreed to participate [14]. Of these of these participants, the included 25 patients with AN (22 girls and 3 boys) present a subgroup recruited between 2016 and 2018. Additionally, 22 HCs (all girls) were recruited and included in the analyses. Table 1 displays the disease characteristics and medication of the patients with AN at the time of hospital admission. Of the 25 patients, one had two, and another patient had four psychiatric comorbidities. Two patients received two medications in parallel. None of the patients took hormonal contraceptives. One patient with AN had been a competitive athlete who developed AN after an injury, and one of the controls was a semi-professional dancer. 

Table 2 shows the patients’ demographic and illness characteristics upon hospital admission and a comparison of PA between patients with AN and HCs. The study groups did not differ significantly in age or height while, expectedly, body weight and BMI, BMI percentile, and %mBMI were significantly lower in patients with AN. While the patients with AN had reported significantly higher premorbid PA1-6 (*p* = 0.017, Table 1, Figure 1) compared with HCs, they took significantly fewer objectively assessed steps per day 21 ± 6 (11–36) days at admission to inpatient treatment (*p* = 0.015; Table 1).

### 3.1. Self-Reported PA over Time

No significant change was reported between PA1-6 and PA-pre (*p* = 0.308), yet there was a more than threefold rise in PA from PA-pre to PA-post onset of AN (*p* = 0.001; Figure 2). Before AN onset, two out of 25 patients (8.0%) exercised for ≥6 h/week, and after AN onset this number increased to 17 of 25 patients (68.0%, *p* < 0.001). After AN onset, 20 of 25 patients (80%) reported the onset of new motivation for PA, and 19 (76%) reported in addition the new onset of high intensity in PA. There was no significant linear relationship between premorbid PA1-6 and its development before and after onset of AN (Table 3).

### 3.2. Associations between PA Parameters 

Table 3 shows the Spearman rank correlation depicting the associations between different PA domains assessed during the interview and objectively with accelerometry. A highly significant association was found between PA1-6 and PA-pre (*p* = 0.001). No significant association was observed between PA1-6 and either PA-post or change of PA-pre to PA-post. Objectively assessed PA at 21 ± 6 (11–36) days after the start of inpatient treatment (steps/d) was not associated with either PA1-6 (*p* = 0.434) or PA-pre (*p* = 0.938). However, significant and direct associations were observed between objectively assessed PA (steps/d) and self-report PA-post (*p* = 0.019) and change of PA-pre to PA-post (*p* = 0.024). 

### 3.3. Relationship between PA Parameters and ED Pathology 

Neither global EDE-Q score nor any subscores were significantly associated with objectively measured steps/d, PA1-6, or PA-pre (Appendix A, Table A1). Conversely, the global EDE-Q score and all subscores were significantly associated with PA-post (except Eating Concern) and change of PA-pre to PA-post.

### 3.4. Differences in PA Patterns between AN Subgroups

Patients with a significant change in PA-pre vs. PA-post (n = 20, 80%) showed greater PA-post (*p* = 0.001), had a higher change in % of PA (*p* = 0.001), and reported significantly more PA-pre (*p* = 0.046) than patients without a significant change of PA. The patients who engaged in ≥6 h of PA/week after AN onset (n = 17, 68%) reported significantly more PA-post (*p* = 0.001), a higher increase of PA in % post AN onset (*p* = 0.004), and significantly more PA-pre (*p* = 0.007) than those who engaged in PA <6 h/week after AN onset. The group that showed new onset of PA motivation (n = 20, 80%) showed significantly more PA-post (*p* = 0.001), had a significant change in % of PA (*p* = 0.001), and reported significantly more PA-pre (*p* = 0.046) than the group without a new onset of PA motivation. The group that showed new onset of high intensity PA (n = 19, 76%) reported significantly more PA-post (*p* = 0.001) and showed a significant increase of PA in % (*p* = 0.001) but had did not report significantly more PA-pre AN (*p* = 0.274). No differences were found for PA1-6 or objectively assessed PA at the beginning of treatment in any of the four group comparisons (with or without significant PA change, PA ≥6 h/week, new onset and/or high intensity of PA, new onset of PA motivation).

### 3.5. Interrater Reliability for Classification of the Patients into PA Subgroups

Interrater reliability was “substantial” for item 1, significant change of PA (Cohen’s Kappa 0.69, *p* = 0.001); “perfect” for item 2, ≥6 h PA/week (Cohen’s Kappa 1.0, *p* = 0.001); and “almost perfect” for item 3, new onset and/or high intensity of PA (Cohen’s Kappa 0.88) and item 4, new onset of PA motivation (Cohen’s Kappa 0.86, all *p* = 0.001). 

### 3.6. Relationship between PA Patterns and Clinical Outcomes

Due to collinearity with the EDE-Q global score detected with Spearman’s rank correlation analysis, all EDE-Q subscales were excluded from the subsequent regression analysis: restraint, eating concern, weight concern, and shape concern (all *p* = 0.001). Detailed results of the univariate and multivariable linear models for change in %mBMI and length of stay are shown in Appendix A, Table A2 and Table A3.

### 3.7. Prediction of Increase in %mBMI and LOS

In the stepwise backwards elimination model, lower admission %mBMI (*p* = 0.001) and new onset PA/high intensity (*p* = 0.004; r^2^ of the model = 0.604) were significant predictors, while neither objectively assessed PA nor ED psychopathology were part of the model. In the stepwise backward elimination model, only higher PA-pre (*p* = 0.003) predicted a longer LOS (r^2^ of the model = 0.368, Table 4).

## 4. Discussion

In this pilot study of the change in PA from before AN to the onset and subsequent inpatient treatment of AN in adolescents, the following main results emerged: (1)Compared with HCs, patients with AN had higher PA during school grades 1-6.(2)Premorbid PA was directly associated with PA before but not with PA after the onset of AN.(3)Using backward stepwise elimination regression, lower admission %mBMI and new onset of high intensity PA were identified as predictors for less increase in %mBMI. Higher PA before AN onset was identified as a predictor for increased LOS.

### 4.1. High Levels of Premorbid PA in Patients with AN and Timing of PA Increase with Respect to Onset of AN

Patients with AN reported spending nearly twice the time doing PA per week (115 vs. 68 min.) during school grades 1-6 compared with HCs. Both the patients with AN and HCs engaged in a broad variety of sports in sports clubs: track and field athletics, basketball, soccer, martial arts, and dance. To our knowledge, only a few previous studies assessed premorbid PA in patients with AN. Davis et al. [10] compared the quantity of premorbid activity in 45 adult patients with AN and BN vs. 19 HCs. To assess premorbid PA, a structured interview was used to recall PA at ages 8, 13, 18, and 23, demonstrating that, on average, patients with AN were almost 100% more physically active at age 13 in min/year and more than 150% physically active at age 18 compared with healthy controls [10]. Our results are in line with these earlier findings [10]. Concomitantly, adult patients with AN with more problematic PA (defined as “time spent exercising exceeded 1 h per day for at least 6 days per week for a period not less than 1 month” and “exercising was described as obsessive, driven, and out of control”) were also more involved in sports at ages 10 and 12 than less-active adult patients with AN [5,6]. A higher number of adolescents with AN who had been highly active as children exhibited more problematic PA with AN during the course of weight loss than those who were average in their activity or less active as children [6,16]. When examining the role of exercise across the lifespan of patients with AN, a “strong premorbid interest” in exercise without quantification or comparison to a control group was described [7]. In our study, the patients with AN reported a shift of PA behaviors after the onset of AN. Many stopped exercising in sports clubs due to various reasons, e.g., physical symptoms, and started to work out at home alone, taking long daily walks while counting their steps or combining running, walking, and biking alone. Thus, prior data and our findings seem to point towards a robust link between early childhood and adolescent exercise behavior and later-occurring AN. 

The timing of the onset of increased PA in patients with AN deserves further attention. In our study, premorbid PA1-6 only correlated significantly with PA before the onset of AN, not with PA after the onset of AN, change of PA (%) after the onset of AN, or objectively assessed PA (steps/d) on admission to inpatient treatment. These findings suggest that high premorbid PA might not automatically be a predictor for increased PA during the course of AN, yet it does not exclude the possibility that premorbid PA might be a risk factor for individuals or a subgroup [23]. In contrast, Davis et al. concluded that a serious commitment to exercise has significance in creating a psychological predisposition to an eating disorder and is a contributing factor in its progression [10]. In addition, Davis proposed that the behavioral synergy when—after onset of AN—strenuous physical activity is combined with starvation, is an integral part of the pathogenesis and the maintenance of the disorder, with the latter being in line with our findings, despite differences between study populations as well as the time and methods of assessment. Finally, in a study examining 20 female adolescents with AN with retrospective questionnaires about the time six months prior to the first hospital admission, Higgins et al. found that significantly increased PA and not decreased energy intake was associated with hospitalization [17].

With these slightly divergent data from human studies, animal experiments that can closely control study set-up and conditions can provide further insight into the development of high PA under food-restricted conditions. Early studies starting in the 1950s in rodents described that experimental animals who had access to food for only one hour/day initially coped well but ran themselves to death once they had access to a running wheel [10,24], a model called activity-based Anorexia (ABA). In 2000, a study showed that leptin suppressed semi-starvation-induced hyperactivity in rats. Leptin-treated, food-restricted rats showed only a gradual increase of PA, while the food-restricted control group showed up to a 300% increase versus baseline PA. Transferring these results to humans suggests that in some cases PA in AN might be biologically driven [25]. In 2012, predictors for ABA were identified by showing differences in susceptibility to ABA in genetically diverse rodent populations. Those mouse strains and rats with high running wheel activity during food restriction exhibited accelerated body weight loss. Baseline running wheel activity (preceding food restriction) strongly predicted ABA susceptibility compared to other baseline parameters. The authors concluded that PA levels may play an important role in PA susceptibility in different rodent species, that the genetic background may play a role, and that premorbid physical activity levels could reflect an early indicator for disease severity [26]. In 2021, an ABA protocol was presented to model both vulnerability and resilience to diet and exercise in female mice [27]. While half of the mice exhibited vulnerability through an expected increase in PA and dramatic weight loss, the other half exhibited “resilience“ to ABA: they adapted to their limited food availability by reducing running wheel activity and increasing food intake, leading to weight stabilization [28].

These findings could support the idea that PA levels may, on a chronological level, precede AN onset, especially in a “vulnerable” subgroup, who during the course of AN exercise more than 20 h/week. 

Davis reported a rise of PA approximately one year prior to the onset of AN, which is in line with our finding that patients of the AN subgroup with a significant change of PA, ≥6 h PA/week, and new onset of motivation for PA showed significantly more exercise before onset of AN than the group without significant change of PA [5,10]. This finding might point to individual, possibly biological traits posing high risk for the development of high PA during the course of AN. 

However, environmental factors may also be at play. For example, Rizk et al. added to the model of development of PA in AN premorbid risk factors, such as having physically active fathers and participation in weight- or esthetics-oriented sports [6].

### 4.2. Association of Increased PA with Clinical Outcome

While predictive factors for clinical outcomes in patients with AN have been intensely researched, self-reported information on premorbid PA or PA change with the onset of AN, to our knowledge, has not been part of systematic investigations [29,30]. In a previous study, we showed that PA patterns, especially objectively assessed high PA in light intensity, had an impact on slowing down inpatient weight gain [14]. In this present study, we show that lower admission percent median BMI and the new onset of high intensity PA might attenuate %mBMI increase during inpatient treatment. In the multivariable analysis of change in body weight during inpatient treatment, the two significant variables, admission %mBMI and new onset PA/high intensity, explained 60% of the variance in change in %mBMI. 

Additionally, higher PA right before the onset of AN might prolong hospitalization. In the multivariable analysis of length of stay during inpatient treatment, PA-pre explained 37% of the variance in length of stay. In a study aimed at synthesizing expert clinical knowledge on defining unhealthy exercise in adolescents with AN, 25 experts reached consensus on 17 items relevant to assess when evaluating adolescents with AN; among these proposed items were duration, intensity, and frequency of exercise as well as motivations or reasons for exercise [31]. This recommendation corresponds with our findings that the interview questions about premorbid PA, PA before the onset of AN, and the new onset of high intensity PA appear to yield information that has clinical relevance with regards to treatment planning to mitigate against slow weight restoration and prolonged inpatient care. Nevertheless, although there has been a shift in the approach toward exercise in the management of patients with AN from disallowing exercise or any relevant PA during the acute treatment period to allowing guided and moderated amounts and types of PA, there is currently no evidence-based international consensus on which patient and illness parameters should be considered when devising a PA or even an exercise plan for patients with AN in the acute, subacute, and maintenance treatment phases. Hence, more research is needed on the most appropriate timing, dose, and type of PA as well as individualized treatment plan development when managing patients with AN during inpatient and outpatient care and in the important relapse prevention phase.

### 4.3. Limitations

The results of this study need to be interpreted within its limitations. First, our research questions required investigation of time periods that extended well into the past, which may have introduced recall bias [10]. In this context, the most reliable method to yield more detailed information appeared to be the semi-structured interview [10,32]. To improve recall accuracy, we used a memory anchor to provide context for the time period being recalled, which might have improved recall quality [33]. Second, objectively assessed PA might be underestimated due to restrictions of the patients with AN in a hospital environment and assessment after consenting procedures, which lead to a delay between admission and actigraphy measurement. Third, participation in the study was voluntary; thus, the composition of the group might have been biased, as patients with AN with certain PA patterns, i.e., low or high PA, might not have wanted to participate. However, the large range of PA patterns measured in the patient group does not point to such a systematic selection bias. Fourth, in our study, the definition of time pre- and post-onset of AN was subjectively recalled by the individuals interviewed. One cannot rule out that psychopathological factors played a role earlier than consciously recognized. Fifth, patients and healthy controls were not matched for sex; there were three male participants in the patient sample, which might have influenced the results. Sixth, the interview was not a validated instrument. Seventh, the sample size of the study was relatively small, which reduced the power of regression analyses, which were exploratory only. Finally, there might be further factors that contribute to the relationship between PA patterns and AN, like perfectionism and difficulties in affect regulation [6], which were not assessed in our study.

## 5. Conclusions

In conclusion, patient-reported PA dimensions are promising predictors of illness course and might offer valuable insight into AN development, pathogenesis, and persistence. In this study, we showed the presence of subgroups with different PA patterns. This exploratory study included relatively few participants; thus, the results are hypothesis-generating. Future studies with larger sample sizes and a longitudinal design are warranted to better understand the role of PA in the evolution of AN and in the response to treatment in adolescents with AN.

## Figures and Tables

**Figure 1 nutrients-16-01889-f001:**
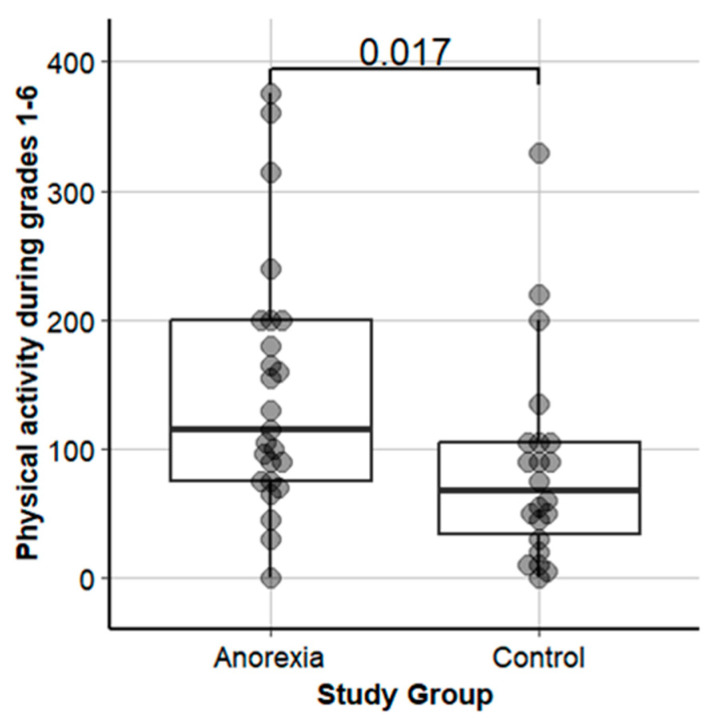
Premorbid physical activity in grades 1-6 in adolescents with anorexia nervosa vs. healthy controls.

**Figure 2 nutrients-16-01889-f002:**
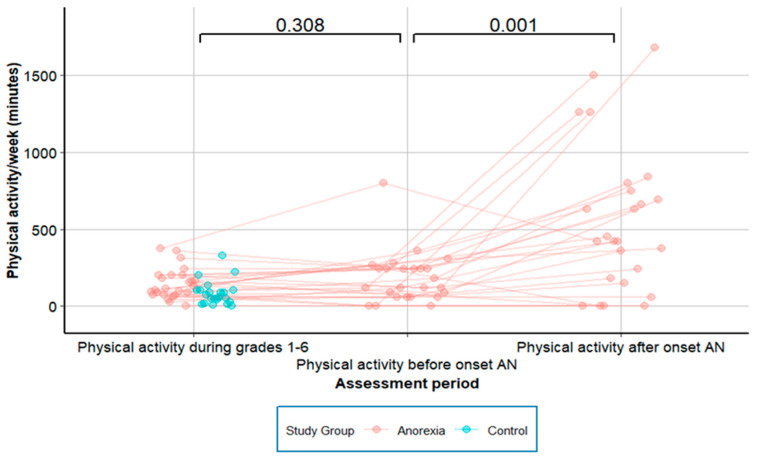
Physical activity in adolescents with anorexia nervosa, longitudinal data.

**Table 1 nutrients-16-01889-t001:** Disease characteristics and psychotropic medications on hospital admission.

	N	Percentage
**Anorexia nervosa subtype**		
Restrictive	16	64.0
Binge–purge	5	20.0
Atypical	4	16.0
**Psychiatric Comorbidities**		
None	14	56.0
Depression	5	20.0
Obsessive compulsive disorder	4	16.0
Anxiety disorder	3	12.0
Borderline Personality Disorder	2	8.0
**Medication**		
None	23	92.0
Stimulating/non-sedating antidepressants	2	8.0
Antipsychotic medication	2	8.0

Antipsychotic medication: Quetiapine, Risperidone; Stimulating/non-sedating antidepressants: Sertraline, Escitalopram.

**Table 2 nutrients-16-01889-t002:** Sociodemographic and clinical characteristics, physical activity, and eating disorder-related psychopathology.

	Patients with AN, Baseline (n = 25)	Healthy Controls (n = 22)	*p*
**Age** (years)	15.1 ± 1.7 [12.1–17.8]	14.7 ± 1.3 [13.0–17.1]	0.494
**Female,** N (%)	22 (88.0%)	22 (100%)	0.004
**Body weight** (kg)	41.2 ± 5.5 [31.3–52.4]	56.2 ± 10.6 [37.1–77.6]	**0.001**
**Height** (cm)	165 ± 8 [150–186]	165 ± 8 [153–182]	0.993
**BMI Percentile** ^1^	2 ± 4 [0–19]	54 ± 29 [3–89]	**0.001**
**%mBMI**	74.8± 6 [65.9–89.5]	102.4 ± 12.1 [78.0–121.2]	**0.001**
**BMI** (kg/m^2^)	15.0 ± 1.0 [13.0–18.0]	20.6 ± 2.7 [15.6–24.0]	**0.001**
**Duration of illness** (months)	10 [0–64]	NA	
**Secondary amenorrhea in females**	17 (77.3%)	0 (0%)	**0.001**
**Primary amenorrhea or No menarche until age 16 ****	3 (13.6%)	4 (18.2%)	1.000
**No amenorrhea in females**	2 (9.1%)	15 (68.2%)	**0.001**
**Hormonal contraception in females**	0 (0%)	3 (13.6%)	0.602
**Steps/d** (admission)	8736 (6755/10,158) [2026–24,536]	11855 (9104/13,954) [4427–23,139]	**0.015**
**PA 1-6 *** (min/week)	115 (75/200) [0–375]	68 (29/105) [0–330]	**0.017**
**PA-pre *** (min/week)	120 (60/240) [0–800]	NA	
**PA-post *** (min/week)	420 (170/767) [0–1680]	NA	
**PA pre-post *** (%)	244 ± 323 [0–1300]		**0.001**
**EDE-Q Global**	3.32 ± 1.69 [0.40–5.40]	NA	
**Restraint**	2.94 ± 1.82 [0.20–5.60]	NA	
**Eating Concern**	2.58 ± 1.72 [0.00–5.60]	NA	
**Weight Concern**	3.50 ± 1.99 [0.00–5.80]	NA	
**Shape Concern**	4.10 ± 1.87 [0.50–6.00]	NA	

Data are expressed as mean ± SD [range] or as median (25th/75th percentile). AN, anorexia nervosa. EDE-Q, Eating Disorder Examination-Questionnaire. PA, physical activity. %mBMI, percent median body mass index. Data on steps available for n = 24 AN and n = 18 HC. Change in PA after onset of AN in % in relation to last reported PA before onset of AN (min/week). NA, not applicable. * self-reported. ^1^ according to Kromeyer–Hausschild. ** In the AN group, 2 girls were below age 16 and one girl had primary amenorrhea; in the HC group, all 4 girls were below age 16; Bold highlights significance.

**Table 3 nutrients-16-01889-t003:** Rank correlations between PA parameters in patients with anorexia nervosa.

	Steps/Day	PA1-6 (min/week)	PA-Pre (min/week)	PA-Post (min/week)	Change of PA-Pre to PA-Post (%)
**Steps/day**	1				
**PA 1-6** (min/week)	0.168 (0.434)	1			
**PA-pre** (min/week)	−0.017 (0.938)	0.633 **(0.001)**	1		
**PA-post** (min/week)	0.476 **(0.019)**	0.284 (0.179)	0.291 (0.168)	1	
**Change of PA-pre to PA-post** (%)	0.46 **(0.024)**	−0.073 (0.735)	−0.154 (0.473)	0.805 **(0.001)**	1

PA, physical activity; PA 1-6, premorbid PA during school grades 1-6; PA-pre, PA before AN onset; PA-post, PA after AN onset; Bold highlights significance.

**Table 4 nutrients-16-01889-t004:** Stepwise backward linear model for change %mBMI (%) and length of stay.

Predictor	Effect Size	Confidence Interval	*p*-Value
**Outcome: change in %mBMI**			
Admission %mBMI (%)	−0.620	[−0.862; −0.378]	0.001
New onset/high intensity PA	5.69	[2.12; 9.25]	0.004
**R^2^ of the model**	**0.604**	**AIC**	**105**
**Outcome: length of stay**			
PA before onset AN	0.149	[0.059; 0.238]	0.003
**R^2^ of the model**	**0.368**	**AIC**	**201**

AN, anorexia nervosa; BMI, Body Mass Index; PA, physical activity; Bold highlights significance.

## Data Availability

Data will be provided upon request to the corresponding author. The data are not publicly available due to privacy.

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
