# Peer review of "The Relationship between Patient Self-Reported, Pre-Morbid Physical Activity and Clinical Outcomes of Inpatient Treatment in Youth with Anorexia Nervosa: A Pilot Study"

_nutrients, 2024, doi:10.3390/nu16121889_

Round 1
Reviewer 1 Report
Comments and Suggestions for Authors
Dear Authors,
I thank the Editor for entrusting me to review this manuscript. Both controlling physical activity and paying attention to eating disorders are particularly important in adolescent children. So congratulations to the authors for taking on the work of finding a link between these issues.
Below are my suggestions / comments:
Table 4:
Unclear Confidence Interval for “Admission %mBMI (%)”. Shouldn't it be “-0.862”?
Completed model coefficients should be interpreted in the text.
Several literature items are published before 2000. It seems that they should be updated.
Lack of a separate conclusions section with detailed conclusions derived from the study.
Author Response
Reviewer: 1
Comments to the Author
I thank the Editor for entrusting me to review this manuscript. Both controlling physical activity and paying attention to eating disorders are particularly important in adolescent children. So congratulations to the authors for taking on the work of finding a link between these issues. Below are my suggestions / comments:
- Table 4: Unclear Confidence Interval for “Admission %mBMI (%)”. Shouldn't it be “-0.862”?
Response: Thank you very much for picking up on this error. You are certainly right and the correct number in the confidence interval for the significant finding of the Admission %mBMI is „-0.862“.
- Completed model coefficients should be interpreted in the text.
Response: Thank you for highlighting this aspect. In the discussion, section 4.2, Association of increased PA with clinical outcome, we have now added the following sentence:
„In the multivariable analysis of change in body weight during inpatient treatment, the two significant variables, admission %mBMI and new onset/high PA, explained 60% of the variance in change in %mBMI.“
Additionally, higher PA right before the onset of AN might prolong hospitalization. We have now added:
„In the multivariable analysis of length of stay during inpatient treatment, PA-pre explained 37% of the variance in length of stay.“
- Several literature items are published before 2000. It seems that they should be updated.
Response: We understand the reviewers concern. We are very sorry that we did not make it clear that the literature published before 2000 are publications of Caroline Davis research group from Canada who first addressed the research question of premorbid physical activity in anorexia nervosa. To our knowledge nobody has built on this body of research since the 1990s. If we missed relevant literature on this specific topic that has been published more recently, we are certainly happy to be pointed to such publications and include them in our manuscript.
- Lack of a separate conclusions section with detailed conclusions derived from the study.
Response: We agree and have added a separate conclusions section:
4.3 Conclusions
“In conclusion, patient-reported PA dimensions are promising predictors of illness course and might offer valuable insight into AN development, pathogenesis and persistence. In this study we showed the presence of subgroups with different PA patterns. This exploratory study included relatively few participants, thus the results are hypothesis-generating. Future studies with larger sample sizes and a longitudinal design are warranted to better understand the role of PA in the evolution of AN and in the response to treatment in adolescents with AN.”
Reviewer 2 Report
Comments and Suggestions for Authors
This manuscript showed great findings, revealing relation between PA and AN. But more in-depth discussion should be added to publish in Nutrients.
Line 98-108: Italics should be modified.
Section 2.1. Why were control patients not fasted while AN patients were fasted?
Section 2.2. Authors should describe more detail information about HC measurement. Did the HC patient also have PA measured between Friday and Sunday? It is a point of concern whether the time subjected to the PA measurement was significantly different from AN patient.
Line 186: Menstrual status was not displayed in table1. Modification should be needed.
Table2: There appears to be a high percentage of amenorrhea in the controls as well, is this appropriate as a control?
The authors discuss the amount of exercise during the PA1-6, PA pre, and PA post phases of AN patients. It is crucial to know what specific exercise they were engaged in. The implications are different between exercising silently alone to lose weight and engaging in group sports for long periods of time (especially in PA1-6 and PApre). A detailed discussion should be added.
It will also be necessary to enrich the discussion on the appropriate amount of exercise, as not exercising at all would probably be problematic.
Comments on the Quality of English LanguageQuality is well.
Author Response
Comments to the Author
This manuscript showed great findings, revealing relation between PA and AN. But more in-depth discussion should be added to publish in Nutrients.
- Line 98-108: Italics should be modified.
Response: Thank you for pointing out this important format detail, that we have corrected as suggested.
- Section 2.1. Why were control patients not fasted while AN patients were fasted?
Response: Thank you for pointing out this aspect that might need clarification. School is mandatory in Germany for grades 1-10, so all appointments for the control group took place in the afternoon. Otherwise the protocol was standardized. We added: „Body weight of the control group subjects was measured during the day (not fasted, because the appointments took place after school) using the same chair scale (MCB300K100M, KERN & Sohn GmbH, Balingen, Germany), and height was measured using the same stadiometer (Seca 220 Stadiometer, Vogel & Halke, Hamburg, Germany).“
- Section 2.2. Authors should describe more detail information about HC measurement. Did the HC patient also have PA measured between Friday and Sunday? It is a point of concern whether the time subjected to the PA measurement was significantly different from AN patient.
Response: Measurements of HCs took place while they stayed in their ambulatory environment. We added: „Otherwise, PA assessment was standardized and took place over a 3-day-period from Friday to Sunday as well.“
- Line 186: Menstrual status was not displayed in table1. Modification should be needed.
Response: Thank you for pointing out this incorrect description of the Table 1 content (menstruation status is displayed in Table 2 instead), which has now been corrected as follows:
„Table 1 displays the disease characteristics and medication of the patients with AN at the time of hospital admission.“
- Table2: There appears to be a high percentage of amenorrhea in the controls as well, is this
appropriate as a control?
Response: We did not use the term „primary amenorrhea“ according to the correct definition. Primary amenorrhea can only be diagnosed if menarche has not occurred after the 16th birthday. All 4 healthy control girls were much younger so the correct title should be „No menarche“. In the group with AN, 2 girls did not have their menarche yet, only one girl had primary amenorrhea.
To avoid confusion, we added the definition of primary amenorrhea to the Methods section as follows:
“Primary amenorrhea was defined as absence of menstruation after the conclusion of the 16th birthday REF), calling absent menstruation before this time point “no menarche until age 16”.
Further, we relabeled the characteristic in the Table as “Primary Amenorrhea or No Menarche until age 16*” and added the following footnote:
“* in the AN group, 2 girls were below age 16 and one girl had primary amenorrhea; in the HC group all 4 girls were below age 16”
- The authors discuss the amount of exercise during the PA1-6, PA pre, and PA post phases of AN patients. It is crucial to know what specific exercise they were engaged in. The implications are different between exercising silently alone to lose weight and engaging in group sports for long periods of time (especially in PA1-6 and PApre). A detailed discussion should be added.
Response: We agree. In the discussion we already stated: „Patients with AN reported spending nearly twice the time with PA per week (115 vs. 68 min.) during school grades 1-6 compared with HCs.“ We have now added:
„Both the patients with AN and HCs engaged in a broad variety of sports in sports clubs: track and field athletics, basketball, soccer, martial arts, dance.“
In line 301 we have further added:
„In our study, the patients with AN reported a shift of PA behaviors after the onset of AN. Many stopped exercising in sports clubs due to various reasons, e.g. physical symptoms, and started to work out at home alone, taking long daily walks while counting their steps or combining running, walking and biking alone.“
- It will also be necessary to enrich the discussion on the appropriate amount of exercise, as not exercising at all would probably be problematic.
Response: Thank you for this remark. We agree that this is a really important aspect but and believe that there should be caution to suggest appropriate amounts of exercise for patients with AN. There is no current consensus in the literature on this topic. Over the last 15 years, a body of evidence has been established that it is safe to incorporate exercise under close supervision in the treatment of AN. In earlier times standard treatment of AN restricted physical activity. To account for this important aspect, we have added the following to the discussion section just before the Limitations section.
“Nevertheless, although there has been a shift in the approach toward exercise in the management of patients with AN from disallowing exercise or any relevant PA during the acute treatment period to allowing guided and moderated amounts and types of PA, there is currently no evidence-based international consensus on which patient and illness parameters should be considered when devising a PA or, even, exercise plan for patients with AN, both in the acute, subacute and maintenance treatment phases. Hence, more research is needed on the most appropriate timing, dose and type of PA, and individualized treatment plan development when managing patients with AN during inpatient and outpatient care and in the important relapse prevention phase.”
Round 2
Reviewer 2 Report
Comments and Suggestions for Authors
The things I pointed out were all well-modified.